# Phase II Trial of the Combination of Alectinib with Bevacizumab in Alectinib Refractory *ALK*-Positive Nonsquamous Non-Small-Cell Lung Cancer (NLCTG1501)

**DOI:** 10.3390/cancers15010204

**Published:** 2022-12-29

**Authors:** Satoshi Watanabe, Kazuko Sakai, Naoya Matsumoto, Jun Koshio, Akira Ishida, Tetsuya Abe, Daisuke Ishikawa, Tomohiro Tanaka, Ami Aoki, Tomosue Kajiwara, Kenichi Koyama, Satoru Miura, Yuka Goto, Tomoki Sekiya, Ryo Suzuki, Kohei Kushiro, Toshiya Fujisaki, Naohiro Yanagimura, Aya Ohtsubo, Satoshi Shoji, Koichiro Nozaki, Yu Saida, Hirohisa Yoshizawa, Kazuto Nishio, Toshiaki Kikuchi

**Affiliations:** 1Department of Respiratory Medicine and Infectious Diseases, Niigata University Graduate School of Medical and Dental Sciences, Niigata 951-8510, Japan; 2Department of Genome Biology, Kindai University Faculty of Medicine, Osaka 589-8511, Japan; 3Department of Respiratory Medicine, Nishiniigata Chuo National Hospital, Niigata 951-8510, Japan; 4Department of Respiratory Medicine, Nagaoka Red Cross Hospital, Nagaoka 940-2085, Japan; 5Department of Respiratory Medicine, Niigata City General Hospital, Niigata 951-8510, Japan; 6Department of Respiratory Medicine, Niigata Prefectural Shibata Hospital, Niigata 951-8510, Japan; 7Department of Respiratory Medicine, Niigata Prefectural Central Hospital, Joetsu 943-0892, Japan; 8Department of Internal Medicine, Niigata Cancer Center Hospital, Niigata 951-8510, Japan; 9Department of Respiratory Medicine, Niigata Medical Center, Niigata 951-8510, Japan

**Keywords:** alectinib, bevacizumab, anaplastic lymphoma kinase (ALK), non-small-cell lung cancer, ctDNA

## Abstract

**Simple Summary:**

Patients with Anaplastic lymphoma kinase (*ALK*)-positive lung cancer after progression of ALK-tyrosine kinase inhibitor have limited treatment options. This study shows clinical efficacy of the combination of alectinib and bevacizumab with acceptable toxicity in patients with *ALK*-positive lung cancer after ALK-TKI failure.

**Abstract:**

Anaplastic lymphoma kinase (*ALK*)-positive lung cancer is a rare cancer that occurs in approximately 5% of non-small-cell lung cancer (NSCLCs) patients. Despite the excellent efficacy of ALK-tyrosine kinase inhibitor in *ALK*-positive NSCLCs, most patients experience resistance. We conducted a phase II study to investigate the combination of alectinib with bevacizumab in *ALK*-positive NSCLC patients after failure of alectinib. In this study, *ALK*-positive nonsquamous NSCLC patients previously treated with alectinib received bevacizumab 15 mg/kg on day 1 every 3 weeks and alectinib 600 mg/day until disease progression. The primary endpoints were progression-free survival (PFS) and the safety of alectinib and bevacizumab. The secondary endpoints included overall survival (OS) and correlation of circulating tumor DNA and plasma proteins with PFS. Of the 12 patients treated, the median PFS was 3.1 months (95% CI 1.2–16.1), and the median OS was 24.1 months (95% CI 8.3-not estimable). The *EML4-ALK* fusion gene in circulating tumor DNA was significantly correlated with shorter PFS (1.2 months vs. 11.4 months, HR 5.2, *p* = 0.0153). Two patients experienced grade 3 adverse events; however, none of the patients required dose reduction. Although the primary endpoint was not met, alectinib combined with bevacizumab showed clinical efficacy in *ALK*-positive patients.

## 1. Introduction

Recent studies have shown that the majority of lung adenocarcinomas are induced by driver gene mutations [1]. Anaplastic lymphoma kinase (*ALK*) gene rearrangement is one of the driver mutation genes that induce non-small-cell lung cancers (NSCLCs) [2]. *ALK* positive lung cancer is a rare cancer that occurs in approximately 5% of non-small-cell lung cancer (NSCLCs) patients. ALK-tyrosine kinase inhibitors (TKIs) have shown survival benefits in patients with *ALK*-positive NSCLCs [3,4,5,6,7,8]. Alectinib is a second-generation ALK-TKI and is highly selective for the ALK fusion protein. Two phase III studies comparing alectinib and crizotinib, which is a first-generation ALK-TKI, in the treatment of naïve *ALK*-positive NSCLCs demonstrated that alectinib significantly improved progression-free survival (PFS) with less toxicity [5,6]. Despite the excellent efficacy of alectinib in *ALK*-positive NSCLCs, almost all patients experience resistance to ALK-TKIs and eventually show disease progression.

Bevacizumab is a monoclonal antibody against vascular endothelial growth factor (VEGF) and has been shown to provide clinical benefits in combination with platinum-based chemotherapy in nonsquamous NSCLC [9]. A previous study demonstrated that the addition of bevacizumab to epidermal growth factor receptor (EGFR)-TKIs increased the overall response rate (ORR) and significantly improved PFS [10]. In addition, a recent phase III study evaluating additional effects of bevacizumab to erlotinib showed significant improvement of PFS in *EGFR* mutation-positive NSCLCs [11]. These results indicate that the addition of bevacizumab to molecular target agents is a promising strategy in patients with NSCLC harboring oncogenic driver genes.

Previous studies indicated that TKI-sensitive clones exist even in lung cancers that have developed acquired resistance to TKIs. *EGFR*-mutated NSCLC remains sensitive to EGFR-TKIs even after radiographic disease progression [12]. Rapid progression of disease after discontinuation of TKIs has been reported in *ALK*-positive and *EGFR*-mutated NSCLC patients [13,14]. The growth of these TKI-sensitive cells accelerated following the discontinuation of TKIs. Although the continuation of TKIs seems to be an effective treatment option beyond progression to avoid rapid disease flare and Ou et al. showed that the continuation of crizotinib beyond disease progression had survival benefits, the effectiveness of the continuation of alectinib in patients who progressed during alectinib treatment remains unclear [15].

Herein, we conducted a phase II study to evaluate the efficacy of the combination treatment of alectinib and bevacizumab in *ALK*-positive nonsquamous NSCLC patients who experienced disease progression during alectinib treatment. Of note, this study explored the biomarkers of the combination of alectinib and bevacizumab with a focus on circulating tumor DNA (ctDNA) and angiogenic proteins.

## 2. Materials and Methods

### 2.1. Patient Population

Eligible patients had histologically or cytologically confirmed stage IIIB/IV NSCLC or postoperative recurrent nonsquamous NSCLC that was *ALK*-positive by fluorescence in situ hybridization (FISH), immunohistochemical assay or reverse transcription PCR and had previously received alectinib and had disease progression. Patients were 20 years of age or older, with an Eastern Cooperative Oncology Group (ECOG) performance status of 0 to 1, a life expectancy of at least 3 months, measurable disease according to Response Evaluation Criteria in Solid Tumors (RECIST) version 1.1, and adequate major organ function. Previous administration of other ALK-TKIs and chemotherapies was allowed. The major exclusion criteria included history or presence of hemoptysis or bloody sputum, any coagulation disorder, and tumor invading or abutting major blood vessels.

### 2.2. Treatment

The patients received bevacizumab 15 mg/kg by intravenous injection on day 1 in a 21-day cycle and alectinib orally twice daily at 600 mg/day, starting from day 1 of cycle 1 until disease progression or the occurrence of an unacceptable adverse event (AE).

### 2.3. Study Design

This phase II study was performed at multiple institutions belonging to the Niigata Lung Cancer Treatment Group (NLCTG). The protocol was approved by the Institutional Review Board of each participating institution, and this study was conducted in accordance with the ethical guidelines of the Declaration of Helsinki. All patients provided written informed consent. The primary endpoints of the study were investigator-assessed PFS and the safety of alectinib and bevacizumab. Secondary endpoints included investigator-assessed ORR, overall survival (OS) and the correlation of ctDNA and angiogenic proteins with PFS.

### 2.4. Clinical Assessments

Tumor response was assessed by the investigators using computed tomography (CT) scans or magnetic resonance imaging according to RECIST v1.1 every two cycles (6 weeks). The severity of toxicity was evaluated using the Common Terminology Criteria for Adverse Events v4.0.

### 2.5. Biomarker Studies

Potential biomarkers in the plasma proteins were evaluated at baseline, 6 weeks, and 12 weeks after the initiation of combination therapy and when disease progressed. Fresh blood samples (14 mL) were collected in EDTA tubes and centrifuged for 10 min at 1400× *g* at room temperature to separate plasma and buffy coats. All samples were stored at −80 °C until the analysis. CtDNA was subsequently extracted from the plasma using an AVENIO cell-free DNA isolation kit (Roche Diagnostics, Mannheim, Germany). Sample plasma analysis was carried out for the following panel of circulating angiogenesis markers and cytokines using Bio-Plex 200 (Bio–Rad, Hercules, CA, USA) and the Human Angiogenesis/Growth Factor Magnetic Bead Panel (Merck Millipore, Burlington, VT, USA): angiopoietin-2, bone morphogenetic protein-9, EGF, endoglin, fibroblast growth factor-2, placental growth factor (PLGF), granulocyte colony-stimulating factor, heparin-binding EGF-like growth factor, endothelin-1, hepatocyte growth factor, follistatin, IL-8, leptin, VEGF-A, VEGF-C, and VEGF-D. The ctDNA samples were analyzed with NextSeq 500 (Illumina, San Diego, CA, USA) and the AVENIO ctDNA Surveillance Kit, which can detect mutations in 197 genes (Roche Diagnostics).

### 2.6. Statistical Analysis

There were no studies investigating the continuation and rechallenge of alectinib beyond progressive disease (PD). If the initial CT evaluation showed PD, the PFS would not be clinically meaningful (the null hypothesis was a ≤2 months PFS). A previous study demonstrated that the median PFS was 1.7 times longer in *EGFR*-mutated NSCLC patients treated with erlotinib and bevacizumab than those receiving erlotinib alone [10]. In addition, another study showed the median duration of erlotinib continuation beyond PD was 3.8 months in patients with NSCLC harboring common *EGFR*-mutations [16]. Based on these findings, the alternative hypothesis was that the PFS was at least 6.4 months. A sample size of 10 achieved 80% power to detect the difference between the null hypothesis median PFS of 2 months and an alternative hypothesis median PFS of 6.4 months. This was at a 10% significance level using a one-sided test based on the elapsed time, assuming a study-follow-up duration of 30 months. PFS was defined as the period from the date of enrollment to the date of the verification of disease progression or death from any cause. OS was calculated from the date of enrollment to the date of death. Kaplan–Meier survival curves were constructed for PFS and OS, and differences between groups were identified using the log-rank test. In plasma protein analysis, differences between groups were assessed using Wilcoxon rank-sum tests. All the reported *p* values were 2-sided, and *p* < 0.05 was considered significant. Statistical analysis was performed using JMP 9.0.2 statistical software (SAS Institute, Cary, NC, USA).

## 3. Results

### 3.1. Patient Characteristics

Twelve patients with a median age of 67 years (range, 30–77) were enrolled between July 2015 and September 2018. The patient characteristics at study entry are shown in Table 1. Ten patients were female, and all patients had adenocarcinoma histology with advanced *ALK*-positive NSCLC, which was determined by FISH, immunohistochemistry and/or reverse transcription PCR. All patients had received alectinib, and half of the patients had been treated with 3 or more regimens before enrollment. At the time of the data cutoff, the median follow-up time for OS was 13.7 months for all patients and 26.4 months for living patients.

### 3.2. Drug Exposure and Tumor Response

All patients were treated with at least one cycle of the combination of alectinib and bevacizumab (median 5; range 1–37). All patients had measurable disease by RECIST. One patient achieved partial response (PR), 7 had stable disease (SD) and 4 had progressive disease (PD), resulting in ORRs and disease control rates (DCRs) of 8% and 67%, respectively.

### 3.3. Efficacy

At the data cutoff (August 31, 2019), all patients had a PFS event, and 6 out of 12 patients had died. As shown in Figure 1, the median PFS was 3.1 months (95% CI 1.2–16.1 and 80% CI 1.5–11.4), and the median OS was 24.1 months (95% CI 8.3-could not be calculated). The treatment details after patient enrollment in the study are shown in Figure 2a. Four patients survived for more than 2 years, and 3 of these patients received the combination of alectinib and bevacizumab for more than 20 cycles. The combination of alectinib and bevacizumab reduced the target lesions in half of all patients during the treatment period (Figure 2b).

### 3.4. Safety

Ten out of 12 patients experienced a drug-related AE, with decreased appetite and proteinuria being the most frequently reported (Table 2). Two patients had a total of 4 types of grade 3 AEs (anemia, diarrhea, hypokalemia and proteinuria). None of the patients required dose reduction; however, one patient needed interruption of alectinib due to bacterial infection.

### 3.5. Biomarker Studies

ctDNA, growth factor and angiogenesis assay data were available from all patients at baseline. The *EML4-ALK* fusion gene was detected in 5 patients (Figure 3a). Three out of these 5 patients had *ALK* mutations (V1180L, L1196M and G1202R) (Figure 3b). The Kaplan–Meier curves for PFS and OS for patients with or without the *EML4-ALK* fusion gene in ctDNA are shown in Figure 3c,d. The median PFS was significantly shorter in patients with the *EML4-ALK* fusion gene than in patients without the *EML4-ALK* fusion gene in ctDNA (1.2 months vs. 11.4 months, HR 5.2, *p* = 0.0153) (Figure 3c). The survival time tended to be shorter among patients with the *EML4-ALK* fusion gene in ctDNA than in patients without the *EML4-ALK* fusion gene (12.5 months vs. not reached, HR 4.6, *p* = 0.0603). The best response in patients with *ALK* gene mutations was SD or PD, and PFS was 5.4 months, 1.2 months and 2.7 months (Patients 1, 2 and 4).

To explore predictive biomarkers of the combination of alectinib and bevacizumab, we compared plasma protein levels between patients with longer (≥10 months, 4 patients) and shorter PFS (<10 months, 8 patients). Although none of the plasma proteins examined in this study were significantly different between the two groups, EGF, VEGF-A and follistatin tended to be higher in patients with shorter PFS, and leptin tended to be lower in patients with shorter PFS (Figure 4a). We next divided the patients into two groups according to the plasma levels of EGF, VEGF-A, follistatin and leptin and compared their PFS. As shown in Figure 4b, patients with high EGF and VEGF-A tended to have shorter PFS; however, patients with high leptin tended to have longer PFS. In contrast, plasma levels of BMP-9 and IL-8 did not affect PFS.

## 4. Discussion

This phase II study aimed to assess the efficacy of the combination treatment of alectinib and bevacizumab in *ALK*-positive nonsquamous NSCLC patients after alectinib treatment. Although the primary objective of the study was not met by PFS, favorable OS and DCR were observed. Eleven patients received alectinib and bevacizumab immediately after alectinib monotherapy, and 4 of these 11 patients had PFS times longer than 6 months. AEs of the combination of alectinib and bevacizumab were generally mild and manageable. Although most patients had been treated with 2–3 regimens before enrollment in this study, all patients received cytotoxic chemotherapies and/or ALK-TKIs after PD on alectinib and bevacizumab. These results suggest that the combination of alectinib and bevacizumab could be a useful treatment option for *ALK*-positive patients whose disease progressed during alectinib treatment.

Currently, crizotinib, ceritinib, alectinib, lorlatinib and brigatinib are available for *ALK*-positive NSCLC patients [3,4,5,6,7,8]. Because alectinib has shown favorable survival benefit with acceptable toxicities in the first-line setting, the antitumor activity of other ALK-TKIs following first-line alectinib has been investigated in *ALK*-positive NSCLC patients with resistance to alectinib. The Ascend-9 study, which investigated ceritinib in alectinib-pretreated *ALK*-positive NSCLC patients, showed that the median PFS was 3.7 months and the ORR was 25% [17]. A global phase 1/2 study reported that *ALK*-positive NSCLC patients treated with lorlatinib after alectinib had a median PFS of 5.5 months in all patients and 9.2 months in Japanese patients [18,19]. A retrospective study evaluating brigatinib after alectinib demonstrated that the ORR was 17% and the median PFS was 4.4 months [20]. Overall, the outcomes of ALK-TKIs in alectinib-refractory *ALK*-positive NSCLC patients are unsatisfactory, and optimal treatment after alectinib is warranted. Although our study showed that the combination of alectinib and bevacizumab was similar to or less effective than ceritinib, lorlatinib and brigatinib, 4 of 12 patients in the current study had long PFS times (Figure 2a). Recently, a phase I/II study that investigated the combination of alectinib and bevacizumab in 11 *ALK*-positive NSCLC patients was reported [21]. This study included 6 ALK-TKI-naïve patients and 6 patients who had received pretreatment ALK-TKIs other than alectinib. Three patients experienced grade 3 drug-related AEs, including proteinuria, hypertension and pneumonitis; however, similar to our study, the combination of alectinib and bevacizumab was well tolerated with no new safety signals.

A previous study suggested several mechanisms of resistance to alectinib [22]. Acquired resistance mechanisms are classified into on-target and off-target mechanisms, and on-target resistance, such as secondary *ALK* mutations, could be predictive factors for patients with alectinib resistance. A previous study demonstrated the presence of different *ALK* mutations and different sensitivities to ALK-TKIs [23]. Because it is difficult to obtain rebiopsy specimens after resistance to ALK-TKIs in some cases, we analyzed ctDNA and plasma protein samples to explore predictive biomarkers for the combination of alectinib and bevacizumab. As shown in Figure 3a, the *ALK* mutations V1180L, L1196M and G1202R were found in 3 patients. The PFS times of these patients were 5.4 months, 1.2 months and 2.7 months, and the overall responses were SD, PD and SD. These findings indicated that patients with secondary *ALK* mutations hardly respond to the combination of alectinib and bevacizumab. Because the PFS and OS times of patients with the *EML4-ALK* fusion gene in ctDNA were shorter than those in patients without the *EML4-ALK* fusion gene, secondary *ALK* mutation and the presence of *EML4-ALK* fusion ctDNA could be negative predictive factors for the combination of alectinib and bevacizumab. Additionally, copy number variations (CNVs) of EGFR and mesenchymal-epithelial transition (MET) may be a negative predictive factor because 6 out of 8 patients with short PFS (<10 months) had CNVs of EGFR or MET, and these CNVs were not found in patients with long PFS. Plasma protein analysis showed that patients with long PFS had low amounts of EGF, VEGF-A and follistatin and high amounts of leptin (Figure 4a). Further analysis revealed that the PFS of patients with low plasma EGF and VEGF-A and high plasma leptin tended to be longer (Figure 4b). The association between plasma VEGF-A and the effect of bevacizumab plus chemotherapy has been reported, but that of EGF, leptin and follistatin [24]. Since our study included a relatively small number of patients and these proteins may be prognostic factors for PFS and OS, further studies are warranted to confirm these findings.

The mechanisms of augmentation of the antitumor effect of alectinib by bevacizumab have not been fully elucidated. Watanabe et al. showed that VEGF-A and VEGFR2 were highly expressed in an *ALK*-altered cell line [25]. They found that the blockade of VEGFR2 suppressed the proliferation of *ALK*-altered cells through inhibition of the oncogenic signaling pathway in vitro. Similar findings were reported in an *EGFR*-mutated NSCLC model [26]. Phosphorylation of extracellular signal-regulated kinase, AKT and signal transducer and activator of transcription 3 were upregulated in erlotinib-refractory tumors. The addition of bevacizumab to erlotinib suppressed the phosphorylation of extracellular signal-regulated kinase and inhibited tumor growth, which was refractory to erlotinib alone.

The current study had several limitations. First, this study included a relatively small number of patients because there were few patients with the *EML4-ALK* fusion gene. Second, biomarkers were investigated using ctDNA, not tumor specimens from lesions progressing during alectinib treatment. There might be a difference between DNA at the tumor site and ctDNA. Third, in our study, bevacizumab was added to alectinib after disease progression. The median PFS of alectinib in the first-line setting was 34.1 months, and there was concern about the increase in AEs associated with long-term treatment with bevacizumab at the time this study was planned [27]. However, the safety and efficacy of first-line treatment with alectinib and bevacizumab was demonstrated in another study [21]. To evaluate the additional effect of bevacizumab on alectinib, it is unclear whether it is better to use alectinib and bevacizumab in alectinib-naïve patients or in patients with alectinib resistance. Fourth, this is a single-arm phase II study, not a randomized phase III study. We cannot obtain definitive conclusions on the effectiveness of the combination of alectinib and bevacizumab.

## 5. Conclusions

To our knowledge, this is the first study to examine the combination of alectinib and bevacizumab in alectinib-refractory patients only. This phase II study suggests that the combination of alectinib and bevacizumab is well tolerated and beneficial for some patients previously treated with alectinib, especially those without the *EML4-ALK* fusion gene in ctDNA.

## Figures and Tables

**Figure 1 cancers-15-00204-f001:**
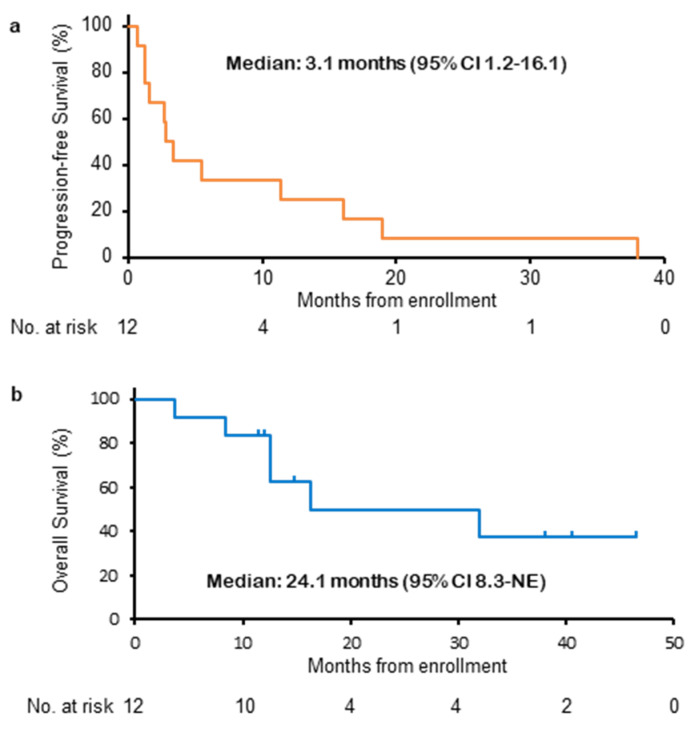
Kaplan–Meier curves for progression-free survival (**a**) and overall survival (**b**). NE; not evaluable.

**Figure 2 cancers-15-00204-f002:**
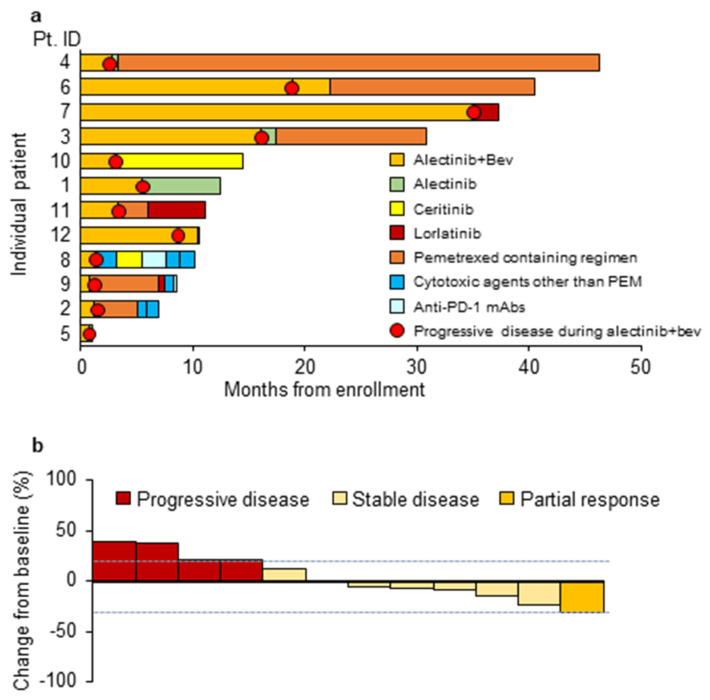
Swimmer and waterfall plots (**a**) Swimmer plot showing duration of response. Each bar represents one patient. (**b**) Waterfall plots of percent change from baseline in measurable tumors at the time of the best response. Pt; patient, PD-1; programmed cell death-1, mAbs; monoclonal antibodies.

**Figure 3 cancers-15-00204-f003:**
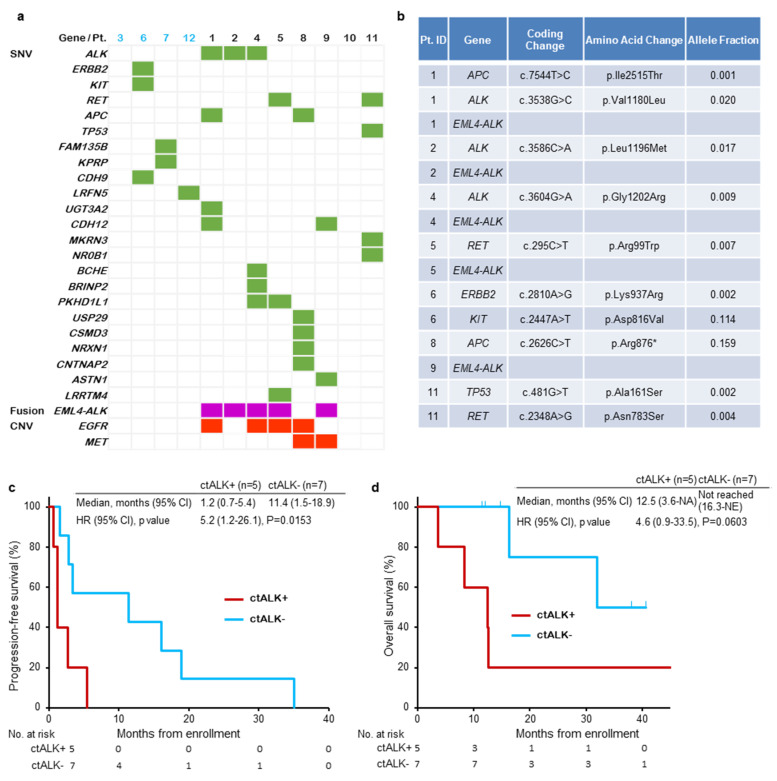
Predictive biomarker analysis using ctDNA. (**a**) Gene mutations and fusion genes detected in individual patients are shown. (**b**) Analysis focused on mutations associated with NSCLC. Kaplan–Meier curves for progression-free survival (**c**) and overall survival (**d**) among patients with or without the *EML4-ALK* fusion gene in ctDNA. SNV; single nucleotide variant, CNV; copy number variant, ctALK; circulating tumor anaplastic lymphoma kinase fusion gene, NE; not evaluable.

**Figure 4 cancers-15-00204-f004:**
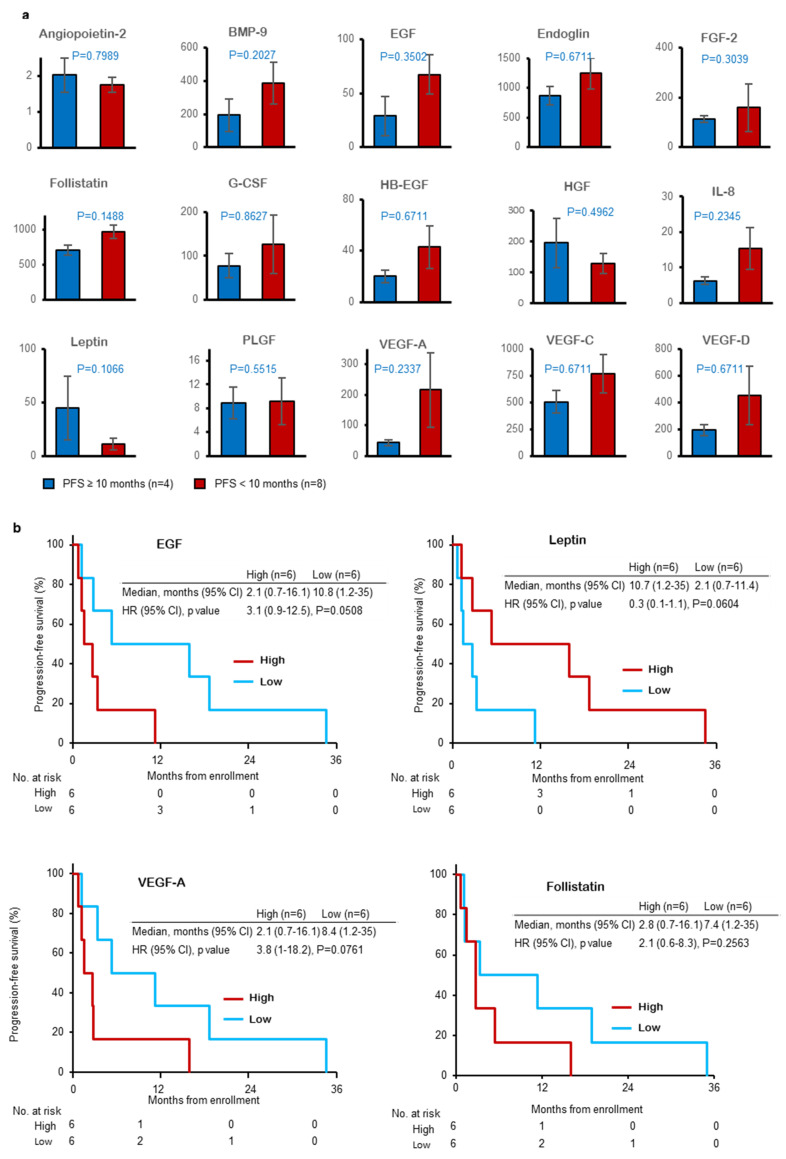
Correlation of plasma protein levels with progression-free survival. (**a**) Levels of plasma protein between patients with longer (≥10 months, 4 patients) and shorter progression-free survival (<10 months, 8 patients). (**b**) Progression-free survival curves of patients with high or low levels of EGF, leptin, VEGF-A and follistatin. BMP-9; bone morphogenetic protein-9, EGF; epidermal growth factor, FGF-2; fibroblast growth factor-2, G-CSF; granulocyte colony stimulating factor, HB-EGF; heparin-binding EGF-like growth factor, PLGF; placental growth factor, VEGF; vascular endothelial growth factor.

**Table 1 cancers-15-00204-t001:** Patient Characteristics.

Characteristics	n = 12	%
Age (y)		
Median (range)	67 (30–77)	
Gender		
Male	2	17
Female	10	83
Performance status		
0/1/2	1/10/1	8/83/8
Histology		
Adenocarcinoma	12	100
Clinical Stage		
Stage IIIB	2	17
Stage IV	10	83
Diagnosis methods		
FISH	11	92
IHC	10	83
RT-PCR	2	17
Previous regimens		
1/2/≥3	3/3/6	25/25/50
Previous ALK-TKI		
Alectinib	12	100
Crizotinib	9	75
Ceritinib	2	17
Best response to prior alectinib		
CR/PR/SD	1/7/4	8/58/33

FISH; fluorescence in situ hybridization, IHC; immunohisto chemistry, RT-PCR; reverse transcription-Polymerase Chain Reaction, ALK-TKI; anaplastic lymphoma kinase-tyrosine kinase inhibitor, CR; complete response, PR; partial response, SD; stable disease.

**Table 2 cancers-15-00204-t002:** Adverse events (n = 12).

	**Grade 1/2**	**Grade** **≥ 3**
Anemia	3	1
Stomatitis	2	
Appetite loss	5	
Nausea/vomiting	1	
Diarrhea	1	1
Constipation	1	
Hypertension	4	
Fatigue	4	
Blood bilirubin increased	1	
AST/ALT elevation	1	
Creatinin elevation	3	
Hyponatremia	1	
Hypokalemia		1
Proteinuria	4	1
Edema	2	
Infection	1	
Pain	2	
Epistaxis	2	
Headache	1	

AST, aspartate aminotransferase; ALT, alanine aminotransferase.

## Data Availability

The data presented in this study are available on request from the corresponding author.

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
