# Peer review of "Phase II Trial of the Combination of Alectinib with Bevacizumab in Alectinib Refractory ALK-Positive Nonsquamous Non-Small-Cell Lung Cancer (NLCTG1501)"

_cancers, 2022, doi:10.3390/cancers15010204_

Round 1

Reviewer 1 Report

Thank you for giving me the opportunity to review your very interesting article. I believe this is a clinically meaningful paper to be appropriate to publish in Cancers. That said, some minor revisions are needed in your current manuscript. Please consider the followings.

Major Comments

1. Article title

Eligible patients of this study are advanced stage NSCLC who are previously treated with alectinib. However, some readers might think that this trial was underwent for untreated ALK+ NSCLC when they see the current article title. You should change the article title below. Phase II trial of the combination of alectinib with bevacizumab in alectinib refractory ALK-positive Nonsquamous Non-Small Cell Lung Cancer (NLCTG1501)

2. Materials and Methods section

Who did judge the PFS and tumor response by RSCIST, by investigator assessment or independent review? You should describe about that.

3. Table 1

You should change the words in Table1. “Best response to alectinib” means the response of alectinib as a prior treatment, right? “Best response of prior alectinib” should better description.

4. Results

Some previous articles have reported TKI efficacy was recovered by long TKI free interval (over 6 or 8 months, or 1 year, the definitions of TKI free interval are different depend on the articles). In your dataset, how was the difference of PFS/ORR according to TKI free interval? I speculate, the population of long TKI free interval is better response and PFS of alectinib+Bev. You should analyze about that.

5. Discussion Section

You described below in Discussion section, “Watanabe et al. showed that VEGF-A and VEGFR2 were highly expressed in an ALK-altered cell line [25].” However, in the reference 25, the authors did not mention about VEGF-A and VEGFR2. And the first author’s name is not Watanabe. Please correct the reference.

6. Conclusion section

You described below in Discussion section. “To our knowledge, this is the first study to examine the combination of alectinib and bevacizumab after alectinib failure.” However, as you mentioned, Lin JJ et al. have already reported phase I/II study to investigate the efficacy and safety of alectinib+BEV including alectinib refractory patients. Your study is not first study. Please correct about that.

Author Response

Thank you for taking the time to review our manuscript. Please find a point-by-point response to each comments below.

Major Comments

  1. Article title

Eligible patients of this study are advanced stage NSCLC who are previously treated with alectinib. However, some readers might think that this trial was underwent for untreated ALK+ NSCLC when they see the current article title. You should change the article title below. Phase II trial of the combination of alectinib with bevacizumab in alectinib refractory ALK-positive Nonsquamous Non-Small Cell Lung Cancer (NLCTG1501)

Response: We thank for the reviewer’s valuable comments. We have changed the manuscript title as suggested.

  1. Materials and Methods section

Who did judge the PFS and tumor response by RSCIST, by investigator assessment or independent review? You should describe about that.

We thank for the reviewer’s comments. The investigators assessed the PFS and tumor responses using RECIST criteria. We have revised the manuscript to include this information.

The following text has been changed in the manuscript:

(Page 3, lines 103-105) The primary endpoints of the study were investigator-assessed PFS and the safety of alectinib and bevacizumab. Secondary endpoints included investigator-assessed ORR, overall survival (OS) and the correlation of ctDNA and angiogenic proteins with PFS.

(Page 3, lines 108) Tumor response was assessed by the investigators using computed tomography scans or magnetic resonance imaging according to RECIST v1.1 every two cycles (6 weeks).

  1. Table 1

You should change the words in Table1. “Best response to alectinib” means the response of alectinib as a prior treatment, right? “Best response of prior alectinib” should better description.

Response: We thank for the reviewer’s valuable suggestion. We have revised Table 1.

  1. Results

Some previous articles have reported TKI efficacy was recovered by long TKI free interval (over 6 or 8 months, or 1 year, the definitions of TKI free interval are different depend on the articles). In your dataset, how was the difference of PFS/ORR according to TKI free interval? I speculate, the population of long TKI free interval is better response and PFS of alectinib+Bev. You should analyze about that.

Response: We thank for the reviewer’s valuable suggestion. In the current study, 11 out of 12 patients were included immediately after alectinib monotherapy. There were no TKI free intervals in these patients. Four of these 11 patients had PFS longer than 6 months, suggesting that the combination of alectinib and bevacizumab could be a useful treatment option for alectinib-refractory ALK-positive patients.

We mentioned these findings on page 11, lines 233-240.

  1. Discussion Section

You described below in Discussion section, “Watanabe et al. showed that VEGF-A and VEGFR2 were highly expressed in an ALK-altered cell line [25].” However, in the reference 25, the authors did not mention about VEGF-A and VEGFR2. And the first author’s name is not Watanabe. Please correct the reference.

Response: We apologize for our mistake. We have changed the reference 25.

  1. Conclusion section

You described below in Discussion section. “To our knowledge, this is the first study to examine the combination of alectinib and bevacizumab after alectinib failure.” However, as you mentioned, Lin JJ et al. have already reported phase I/II study to investigate the efficacy and safety of alectinib+BEV including alectinib refractory patients. Your study is not first study. Please correct about that.

Response: Thank you for your valuable comments. The clinical study reported by Lin JJ included both alectinib-naïve and alectinib-refractory patients. In contrast, the current study included only alectinib-refractory patients. To reflect this information, we have revised the manuscript.

The following text has been changed in the manuscript:

(page 12, lines 313-314) To our knowledge, this is the first study to examine the combination of alectinib and bevacizumab in alectinib-refractory patients only.

Reviewer 2 Report

Manuscript cancers-2110095 by Watanabe et al. describes the results of a phase II clinical trial study using combination therapy of alectinib and bevacizumab to treat ALK positive NSCLC patients after manifestation of alectinib resistance. Authors evaluate PFS and OS in all patients, side effects, best response, predictive biomarker based on ctDNA and plasma protein analysis. Authors concluded that the combination therapy was not satisfactory overall but did show extended PFS in part of patients.

This is a well written and organized manuscript. I only have minor comments:

 it is recommended to include the full term in first place when introduce acronyms. e.g. term PD appeared first time in line 128 but the full term appeared in line 160-161. In line 274, no full term associated with MET.

3.3 Efficacy: in lines 169-171, authors described combination therapy reduced target lesions by 50% of all patients during the treatment period. However, Figure 2b didn't reflect 50% reduction for any patient.

3.5 Biomarker studies: what about the difference of PFS with high and low levels of BMP-9 and IL-8 that also show low p value and potential predictive biomarker?

Author Response

Thank you for taking the time to review our manuscript. Please find a point-by-point response to each of comments below.

Manuscript cancers-2110095 by Watanabe et al. describes the results of a phase II clinical trial study using combination therapy of alectinib and bevacizumab to treat ALK positive NSCLC patients after manifestation of alectinib resistance. Authors evaluate PFS and OS in all patients, side effects, best response, predictive biomarker based on ctDNA and plasma protein analysis. Authors concluded that the combination therapy was not satisfactory overall but did show extended PFS in part of patients.

This is a well written and organized manuscript. I only have minor comments:

 it is recommended to include the full term in first place when introduce acronyms. e.g. term PD appeared first time in line 128 but the full term appeared in line 160-161. In line 274, no full term associated with MET.

Response: We thank for the reviewer’s comments. We have added the full terms.

The following text has been changed in the manuscript:

(page 3 line 108) computed tomography (CT)

(page 3 line 130) progressive disease (PD)

(page 11 lines 277-278) mesenchymal-epithelial transition (MET)

3.3 Efficacy: in lines 169-171, authors described combination therapy reduced target lesions by 50% of all patients during the treatment period. However, Figure 2b didn't reflect 50% reduction for any patient.

Response: We thank for the reviewer’s comments. This sentence means the combination of alectinib and bevacizumab reduced the target lesions in half of patients. To make this clear, we have revised the manuscript.

The following text has been changed in the manuscript:

(page 5 lines 171-173) The combination of alectinib and bevacizumab reduced the target lesions in half of all patients during the treatment period (Figure 2b).

3.5 Biomarker studies: what about the difference of PFS with high and low levels of BMP-9 and IL-8 that also show low p value and potential predictive biomarker?

Response: We evaluated whether all of proteins presented in figure 4a could be potential predictive biomarkers. Plasma levels of BMP-9 and IL-8 did not affect PFS (P=0.5077 and P=0.4801). To include these findings in the result section, we have revised the manuscript.

The following text has been changed in the manuscript:

(page 7 lines 211-212) In contrast, plasma levels of BMP-9 and IL-8 did not affect PFS (data not shown).

Reviewer 3 Report

This is a well-designed phase II study of alectinib in combination with bevacizumab in ALK TKI resistant NSCLC. Acceptance as the current status is my recommendation.

Author Response

This is a well-designed phase II study of alectinib in combination with bevacizumab in ALK TKI resistant NSCLC. Acceptance as the current status is my recommendation.

Response: Thank you for taking the time to review our manuscript.